

# Expression of the "4.2 ka event" drought in the southern Rocky Mountains, USA

David T. Liefert[1], Bryan N. Shuman[1]

1) Department of Geology and Geophysics, University of Wyoming, Laramie, WY 82070, USA

*Correspondence to*: David T. Liefert (dliefert@openspace.org)



**Abstract**
The use of the climatic anomaly known as the "4.2 ka event" as the stratigraphic division
between the mid- and late Holocene has prompted debate over its impact, geographic pattern,
and significance. The anomaly has primarily been described as abrupt drying, but evidence of
hydroclimate change at ca. 4 ka is inconsistent among sites globally, and few sites in North
America document a major drought. Climate records from the southern Rocky Mountains
demonstrate the challenge with diagnosing the extent and severity of the anomaly. Dune-field
chronologies and a pollen record in southeast Wyoming reveal several centuries of low moisture
at around 4.2 ka and prominent low stands in lakes in Colorado suggest the drought was unique
amid Holocene variability, but detailed carbonate oxygen isotope ($\delta^{18}O_{carb}$) records from
Colorado do not record it. We find new evidence from $\delta^{18}O_{carb}$ in a small mountain lake in
southeast Wyoming of an abrupt reduction in effective moisture or snowpack from
approximately 4.2–4 ka that coincides in time with the other evidence from the southern Rocky
Mountains and the western Great Plains of regional drying at around 4.2 ka. We find that the
$\delta^{18}O_{carb}$ in our record may reflect cool-season inputs into the lake, which do not appear to track
the strong enrichment of heavy oxygen by evaporation during summer months today. The
modern relationship differs from some widely applied conceptual models of lake-isotope systems
and may indicate reduced winter precipitation rather than enhanced evaporation at ca. 4.2 ka.
Inconsistencies among the North American records, particularly in $\delta^{18}O_{carb}$ trends, thus show that
site-specific factors can prevent identification of the patterns of multi-century drought. However,
the prominence of the drought at ca. 4 ka among a growing number of sites in the North
American interior suggests it was a regionally substantial climate event amid other Holocene
variability.



## 1. Introduction


Rapid climate changes are well documented in the late Pleistocene and early Holocene,
such as during the Younger Dryas chronozone and at 8.2 ka (thousands of years before present;
Alley et al., 1997; Clark et al., 1999; Von Grafenstein et al., 1998), but mid- to late-Holocene
changes are less well understood (Wanner et al., 2008, 2011). One potential abrupt change
during this time, a multi-century climatic anomaly known as the "4.2 ka event," has been used
as the benchmark for the stratigraphic division between the mid- and late Holocene (Walker et
al., 2019). Consequently, the 4.2 ka event has become a topic of scrutiny with debate over its
impact, geographic pattern, and significance (Bradley & Bakke, 2019; Weiss, 2016, 2019). The
ostensibly global event has primarily been described as a dry episode at low and mid-latitudes
(Booth et al., 2005; Nakamura et al., 2016; Di Rita & Magri, 2019; Scuderi et al., 2019; Xiao et
al., 2018). Consistent with spatial variation expected from climate variability that shifts
atmospheric waves and dynamics, however, some regions show increased precipitation (Huang
et al., 2011; Railsback et al., 2018) or no change (Roland et al., 2014).
Despite the widespread examination of the 4.2 ka event, its cause and significance amid
other millennial-to-centennial climate variability during the Holocene remain unknown. Recent
simulations have produced similar patterns of extended drought in the northern hemisphere
without external forcings such as insolation changes or volcanism (Yan & Liu, 2019), and others
confirm that multi-decadal megadroughts can arise through internal climate variability without
changes in boundary conditions (Ault et al., 2018). Internal climate dynamics and feedbacks
could also interact with stochastic variability and external forcing to produce such events without
consistent or linear relationships to the forcing; forcing may only have a modest probability of



triggering rapid climate changes (Renssen et al., 2006). Less clear is how unusual or frequent
prolonged 'megadroughts' may be within the Holocene across different regions.

That such droughts can occur stochastically indicates the 4.2 ka event could be an

example of typical late-Holocene climate variability at multi-century time scales (Shuman &
Burrell, 2017), but in at least some regions, the event may be exceptional within the spectrum of
Holocene variability. Evidence for a major hydroclimate change at ca. 4 ka has been growing in
the North American midcontinent (Booth et al., 2005; Carter et al., 2018; Dean, 1997; Denniston
et al., 1992; Halfen & Johnson, 2013; Jiménez-Moreno et al., 2019), and adjacent regions, such
as the northeastern United States, where it is recorded as part of a series of Holocene wetting and
drying events (Newby et al., 2014; Shuman et al., 2019; Shuman & Burrell, 2017). The event's
significance or uniqueness has been difficult to verify, however, in North America where few
sites document the anomaly compared to other regions of the mid-latitudes globally (Ran &
Chen, 2019; Zhang et al., 2018).

Records from the southern Rocky Mountains demonstrate the challenge. In the mid-

latitude Rocky Mountains, only dune and pollen records have been explicitly interpreted to show
the 4.2 ka event. Initial recognition in North America derived from the timing of the reactivation
of the Ferris, Seminoe, and Casper Dune Fields in east-central Wyoming (Fig. 1; Booth et al.,
2005; Halfen et al., 2010; Stokes & Gaylord, 1993), but the extent of the drought has been
unclear because other dune-field chronologies in the adjacent western Great Plains do not clearly
document the drought (Dean, 1997; Halfen & Johnson, 2013; Mason et al., 1997). More recently,
Carter et al. (2013, 2017a, 2018) used fossil pollen from Long Lake in the Medicine Bow
Mountains, south of the Wyoming dune fields (Fig. 1), to identify a 150-year interval of
increased temperature and decreased precipitation centered at 4.2 ka. The inferred precipitation



reductions were largest in springtime (Carter et al., 2018), when snowfall in the southern Rocky
Mountains is highest today (Mock, 1996). Consistent with this interpretation, prominent
stratigraphic evidence of lake-level changes in Colorado and Wyoming lakes could indicate that
low water phases at ca. 4.2 ka were one of the most prominent hydrologic changes during the
Holocene (Jiménez-Moreno et al., 2019; Shuman et al., 2009; Shuman et al., 2014, 2015). It
stands out as one of the only multi-centennial features in a summary of low lakes in the Rocky
Mountains during the late-Quaternary (Shuman and Serravezza 2017).

By contrast, the 4.2 ka event does not appear in stable oxygen isotope records from lakes

in the same region, such as detailed carbonate-$\delta^{18}$O ($\delta^{18}O_{carb}$) records from Bison and Yellow
lakes, Colorado (Fig. 1; Anderson, 2011, 2012). Widely applied conceptual models of lake-
isotope systems indicate that hydrologic controls on isotope budgets and the timing of carbonate
formation should play an important role in how the event was recorded, but that the isotopic
response should vary predictably by hydrologic setting (e.g., Anderson et al., 2016; Leng &
Marshall, 2004; Talbot, 1990). According to such models, long lake-water residence times and
high rates of evaporation cause hydrologically closed lakes (i.e., terminal basins) to record shifts
in effective moisture (precipitation – evaporation) because endogenic carbonates will typically
precipitate in evaporated, $^{18}$O-rich water during the warm summer months. Drought could affect
such a lake-isotope system by both increasing evaporation and changing seasonal precipitation,
such as by reducing snowpack. In hydrologically open lakes with short residence times, the
continual replacement of evaporated water creates isotopic sensitivity primarily to the seasonal
balance of precipitation without a strong evaporation effect. Many lakes fall somewhere between
fully hydrologically open and closed and additional site-specific influences may also override
such expectations. Consequently, not all stable oxygen isotope records from lakes may have been





sensitive to the specific climate variables that changed at 4.2 ka. Modern lake-water
measurements can help to identify the relative influences of different controls (Fig. 2; Anderson
et al., 2016).

Here we present a new $\delta^{18}O_{carb}$ record from Highway 130 Lake (HL) in southeast

Wyoming near where other Holocene paleohydrological and paleoecological records have been
developed (Mensing et al., 2012; Minckley et al., 2012; Brunelle et al., 2013). HL is an
intermittently closed subalpine lake in the Medicine Bow Mountains, within 20 km of Long Lake
where fossil pollen indicates a prolonged 'megadrought' at 4.2 ka (Fig. 1; Carter et al., 2018).
The lake is also <60 km from Upper Big Creek Lake, Colorado, where a prominent
paleoshoreline detected in geophysical surveys and cores indicates low water after 4.7 ka (Fig. 1;
Shuman et al., 2015). Previous work at HL indicates a strong influence of evaporation on the
lake and its water isotopes, which we compare with Bison and Yellow lakes in Colorado (Fig. 2;
Liefert et al., 2018). We discuss how dissimilarities in $\delta^{18}O_{carb}$ among lakes, possibly driven by
non-climatic factors, could complicate interpretations of the patterns of past hydroclimate
changes including megadroughts and Holocene trends. Together these outcomes may clarify the
timescales on which drought operates within a critical headwater area of North America, but also
confirm that interpretations of stable isotope records of past hydroclimate changes may depend
heavily on site-specific dynamics.

2.  **Site description**

HL (41°21'05" N, 106°15'50" W; 3,199 m a.s.l. (above sea level)) fills a shallow

depression in the uneven terrain covering the Libby Creek watershed (12 km² surface area) in the
Snowy Range, a southwest trending subsection of the Medicine Bow Mountains in southeast



Wyoming (Fig. 1). Around HL, subalpine coniferous forests interspersed with open meadows
grow on thin glaciated soils and tills between the frequent outcroppings of the underlying
siliceous metadolomite (Houston & Karlstrom, 1992; Musselman et al., 1992). Southeast
Wyoming has a semi-arid climate, but high elevations in the Medicine Bow Mountains receive
about 1,000 mm of precipitation each year, with approximately 70% of annual totals falling as
snow from October to June (Mock, 1996). Local average wind speeds are high (~5 m/s) and
minimum winter and maximum summer temperatures typically reach -23°C and 21°C,
respectively.

The surface watershed around HL occupies ~0.45 km$^2$, while the lake has a surface area

of ~0.02 km$^2$, a maximum (spring) water depth of ~200 cm, and declines in water level by ~30
cm from July to late October (Liefert et al., 2018). Ice covers HL from approximately October to
May and stream connections shut off in June following spring flooding. Measurements reveal no
thermal stratification because of the shallow water depth, flat-bottom bathymetry, and high
average wind speeds, which promote mixing throughout the water column (Bello & Smith, 1990;
Stewart & Rouse, 1976). Liefert et al. (2018) found that evaporation could account for as much
as 83% of the seasonal water loss at HL, though the stable water level and temperature compared
to nearby lakes of similar size and depth indicates shallow groundwater flow-through driven by
seasonal precipitation and infiltration (Rautio & Korkka-Niemi, 2011; Rosenberry & LaBaugh,

2008).


3.  **Methods**

To measure the modern oxygen and hydrogen isotope compositions of the lake water

($\delta^{18}O$ and $\delta D$, respectively) and specific conductance, water samples were collected at





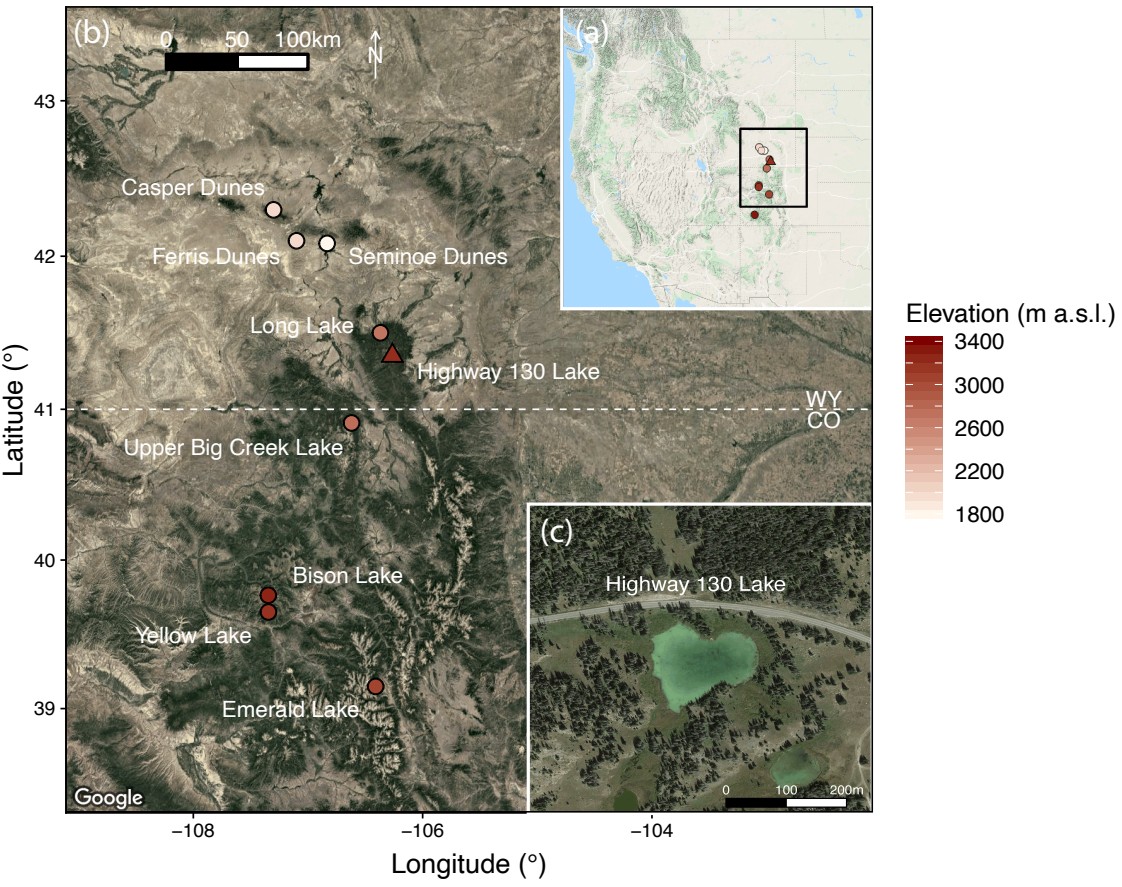

**Figure 1.** Locations of study site and related climate records. (**a**), Highway 130 Lake (triangle) and related

climate records (circles) lie within the southern Rocky Mountains, a critical headwater area in the western

United States that contributes snowmelt to the Colorado and North Platte Rivers. (**b**), Study site locations

in the Colorado Front Range and Medicine Bow Mountains, southeast Wyoming. Little Molas Lake, a

comparator site in the inset map (a), lies south of the focal region (b). (**c**), Highway 130 Lake lies within

the Snowy Range, a subsection of the Medicine Bow Mountains. Google images (© Google Maps 2021)

were acquired using the ggmap package in R (Kahle & Wickham, 2013).

approximately biweekly intervals from June to October in 2017. Additional samples of snowfall,
snowpack, rain, and groundwater (from springs and wells) were collected to measure the range in



water isotope values of the watershed's hydrologic components. Isotopic ratios were measured at
the University of Wyoming Stable Isotope Facility using a Picarro L2130-I Cavity Ring Down
Spectrometer and specific conductance was measured using a YSI Multiparameter Water Quality
Meter. We acquired meteorological data from SNOTEL stations near HL at Brooklyn Lake,
Wyoming (ID 367; 3,121 m a.s.l.; 41.36 °N, -106.23 °W), and at Bison Lake, Colorado (ID 345;
3,316 m a.s.l.; 39.76 °N, -107.36 °W), to compare the modern ratios of snow/rain that control the
seasonal balance of precipitation at the lakes.

In October 2016 we installed a pressure transducer (Onset HOBO U20 Level Data

Logger) to measure the water level of HL at 30-min intervals; freezing conditions required that
we secure the transducer to the lakebed inside a bladder filled with antifreeze. To compensate for
barometric pressure changes we adjusted the transducer data using pressure measurements from
the nearby Glacier Lakes Ecosystem Experiments Site Brooklyn Tower Ameriflux site (GLEES
Tower; US-GLE: https://ameriflux.lbl.gov/sites/siteinfo/US-GLE; 41°21'57" N, 106°14'23" W;
3,191 m a.s.l.). In late January 2017, we installed a conductivity data logger (Onset HOBO U24
Conductivity Data Logger) at the same location and water depth as the pressure transducer to
measure the range in conductivity (converted to specific conductance at 25 °C) at 30-min
intervals of the unfrozen water underlying the ice cover to examine the seasonal patterns of water
chemistry that influence carbonate formation.

At the same time, we collected a 70-mm diameter sediment core with a modified

Livingston piston corer from the center of HL where the combined water and ice depth reached
approximately 90 cm; we used this depth to calibrate the pressure transducer. The organic and
carbonate content of contiguous 1-cm intervals of the sediment core were measured by weighing
the residual sediment after burning the samples at 550 and 1000 °C, respectively. One-cm$^3$ sub-



samples were isolated from each interval after the 550 °C burn for isotopic analysis to remove
organic matter; comparison with organic removal using oxidizing agents indicated no additional
fractionation. The sub-samples were also sieved using a 63-μm mesh to isolate the fine fraction
for isotopic analysis using a Thermo Gasbench coupled to a Thermo Delta Plus XL isotope ratio
mass spectrometer at the University of Wyoming Stable Isotope Facility. X-ray powder
diffraction confirmed that the samples contained only calcite. Ostracod tests were present in less
than 10 of the 300 samples. We report $\delta^{18}O_{carb}$ in the per mil (‰) notation relative to the Vienna
Pee Dee Belemnite (VPDB) standard.

We isolated sedimentary charcoal (>125 μm) and conifer needles from the sediment core

for radiocarbon analyses to estimate sedimentation rates and oxygen isotope chronology
calibrated to radiocarbon years using intcal13 (Reimer et al., 2013) and the age-depth model was
generated using Bchron (Parnell et al., 2008). Radiocarbon samples were analyzed at the
University of California Irvine Keck Carbon Cycle facility.

4.  **Results**
**4.1 *Modern water-chemistry and level measurements***

Lake-water $\delta^{18}O$ and $\delta D$ in HL increased during the ice-free season from -17.8‰ and -

132‰ (sampled in late June) to -10.8‰ and -94.2‰ (sampled in late October), respectively
(black circles, Fig. 2). The local evaporation line (LEL) defined by the HL samples (thick black
line, Fig. 2) traces the LEL defined by samples from lakes in the Colorado Front Range (red
dashed line, Fig. 2; Henderson & Shuman, 2009). Several consecutive years of measurements
reveal that isotope values at HL are consistent from year to year. The LEL's deviation from both
the global meteoric water line (GMWL; Fig. 2) and isotope composition of the hydrologic inputs


(open symbols, Fig. 2) indicates a strong evaporative influence. $\delta^{18}O$ and $\delta D$ values at HL also
indicate stronger fractionation by evaporation compared to representative warm-season isotope
compositions measured at Bison and Yellow lakes (thick purple and yellow lines, Fig. 2;
Anderson, 2011, 2012), which remained closer to the composition of meteoric waters. Longer
lake-water residence time and higher evaporation in HL thus appear to produce a greater range of
warm-season isotope compositions compared to Bison and Yellow Lakes.

The different lake-water-$\delta^{18}O$ values among the lakes contrasts with their similar

seasonal precipitation patterns. The modern ratio of snow/rain, which can determine the mean
precipitation and lake-water $\delta^{18}O$, is comparable in the watersheds of HL and Bison Lake (inset
plot, Fig. 2). Other modern differences among the lakes, which all have surface areas of < 0.1
km$^2$, include that the maximum water depth of HL is several meters shallower than Bison and
Yellow Lakes (Anderson, 2012) and that the summer lake-water temperatures in HL typically
range from 8–12 °C, which is cooler than the epilimnion at Yellow Lake (Anderson, 2012). HL
is also several degrees cooler than nearby lakes also without thermal stratification (Liefert et al.,

2018).

Continuous measurements of specific conductance began in early February when the

combined water and ice depth reached approximately 90 cm (Fig. 3). Specific conductance
increased from 700 µS/cm to 1,115 µS/cm by early April while the lake surface was frozen. The
specific conductance fell below 500 µS/cm as the lake flooded with snowmelt in early May.
Specific conductance ranged from 250–300 µS/cm after the conductivity data logger was
removed in late June and before the lake froze over in the fall, and the water depth stayed
between 100–150 cm, which was low compared to previous years.



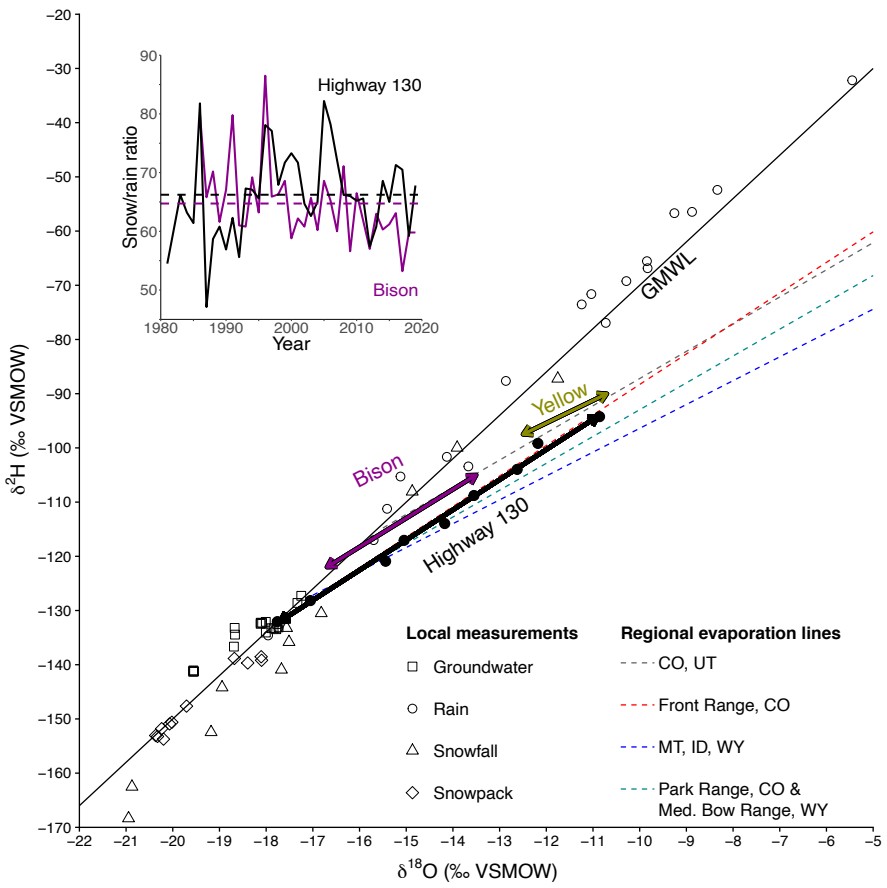


**Figure 2.** Modern measurements of $\delta^{18}$O and $\delta$D. Regional evaporation lines (dashed lines) intersect the global meteoric water line and represent the linear regression of lake-water isotope compositions in a region (Henderson & Shuman, 2009; Anderson et al., 2016). Isotopic measurements from the study watershed (open symbols) show the range in isotope compositions of hydrologic inputs to Highway 130 Lake from the watershed. Arrows represent the range in modern $\delta^{18}$O and $\delta$D values of Highway 130 Lake (black), Bison Lake (purple; Anderson, 2011), and Yellow Lake (yellow; Anderson, 2012) throughout the ice-free season, and black dots show the individual measurements at Highway 130 Lake. The inset plot shows the modern annual ratio of snow/rain for the SNOTEL stations nearest Highway 130 Lake (black line) and Bison Lake (purple line) and the dashed lines show the means.



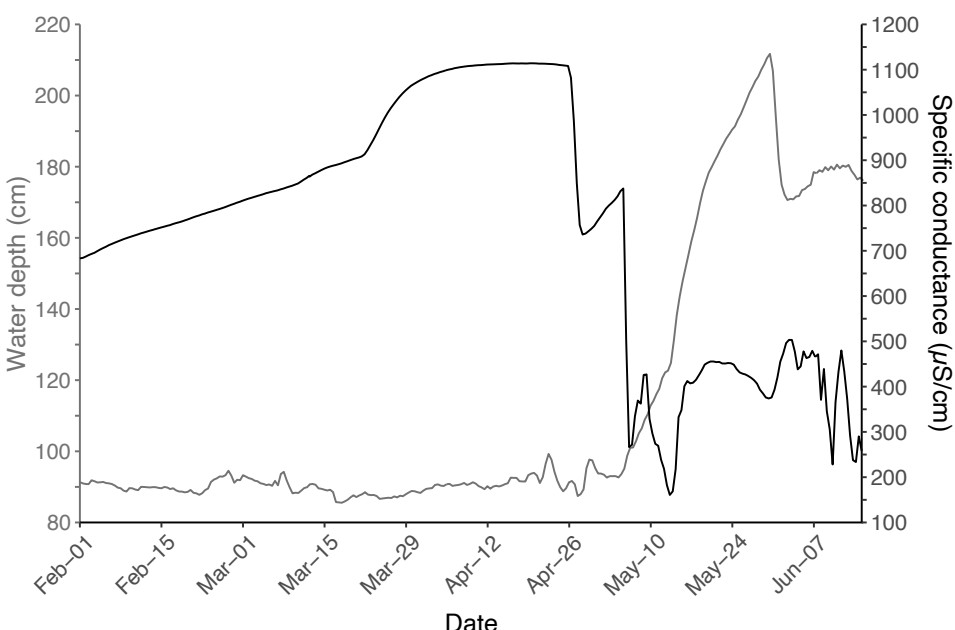

**Figure 3.** Measurements of water depth (gray line) and specific conductance (black line) at Highway 130

Lake in 2017.



**4.2 *Sediment characteristics***

The 333-cm core from HL extends to at least the early Holocene and contains

predominantly carbonate sediment underlain by silicate clays (Fig. 4). The upper 303 cm
contains from 5–55% in organics and 5–90% in carbonate; the core above the basal 30 cm has a
mean carbonate content of 65%. In the basal unit, the carbonate content drops below 5%, which
was too low for isotopic analysis. The age-depth model (black line with 2-sigma gray uncertainty
band, Fig. 4, Table 1) reveals average net sediment accumulation rates of 18 cm/kyr (thousand
years) from 11.7–4.4 ka and 45.5 cm/kyr from 4.4 ka to present. High rates of net sedimentation





correspond with intervals of high carbonate flux into the lake, indicating that carbonate
production may largely control sedimentation rates. The carbonate content and carbonate flux,
representing the mass of carbonates deposited per unit area per year, increased simultaneously
with the sedimentation rate at 4.4 ka (Fig. 4), but the percent carbonate content subsequently
declined until 4.0 ka. The radiocarbon age at 119-cm depth ($3.072 \pm 0.03$ ka) has an age similar
to the date at 67-cm depth ($3.031 \pm 0.02$ ka), which may indicate a reworked upper age (black
dots in Fig. 4). However, high total sediment and carbonate accumulation rates are inferred even
if the upper age was excluded from the age-depth model.


Table 1. Calibrated radiocarbon ages used for the age-depth model.

| Lake | Core | Depth (cm) | Material | Lab number | Age ($^{14}$C yr BP) | Uncertainty ($1 \sigma,^{14}$C yr BP) | Calibrated age ranges ($1 \sigma$, cal yr BP) | | |
|---|---|---|---|---|---|---|---|---|---|
| | | | | | | | Median | Maximum | Minimum |
| Highway 130 Lake | 2A | 18 | Charcoal | UCIAMS-194167 | 850 | 30 | 748 | 783 | 726 |
| | | 67 | Charcoal | UCIAMS-194168 | 2,900 | 20 | 3,033 | 3,070 | 2,996 |
| | 2B | 119-121 | Charcoal | UCIAMS-194169 | 2,925 | 15 | 3,073 | 3,144 | 3,004 |
| | | 154-156 | Charcoal | UCIAMS-194170 | 3,660 | 35 | 3,986 | 4,081 | 3,921 |
| | | 193-195 | Charcoal | UCIAMS-194171 | 3,840 | 20 | 4,241 | 4,290 | 4,157 |
| | | 204 | Charcoal | UCIAMS-194172 | 3,965 | 20 | 4,438 | 4,508 | 4,412 |
| | | 239 | Charcoal | UCIAMS-194173 | 6,210 | 60 | 7,096 | 7,132 | 7,007 |
| | | 302 | Charcoal | UCIAMS-194174 | 9,580 | 25 | 10,927 | 11,074 | 10,781 |


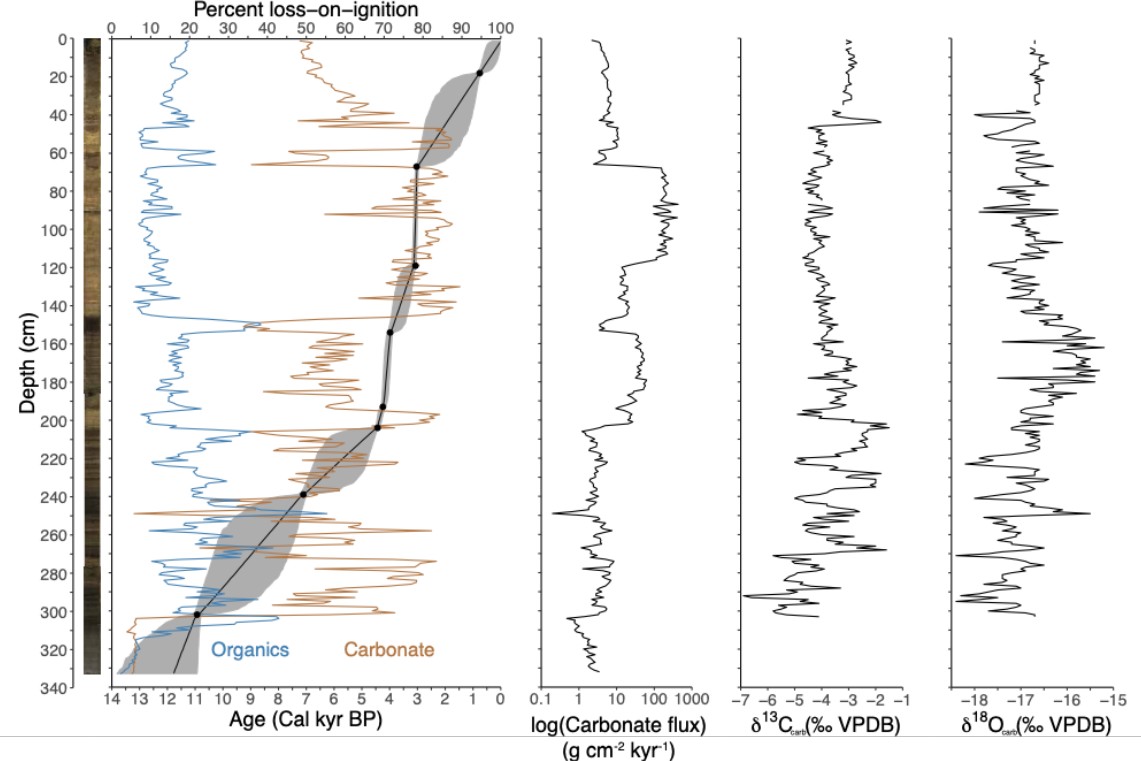

**Figure 4.** Percent organics, percent carbonate, carbonate flux, $\delta^{13}C_{carb}$, and $\delta^{18}O_{carb}$ are shown by depth

alongside an image of the 333-cm-long sediment core from Highway 130 Lake. Radiocarbon ages (black

dots) were used to create the age-depth model and gray uncertainty band (2 sigma).


### 4.3 Sedimentary oxygen and carbon isotopes

$\delta^{13}C_{carb}$ and $\delta^{18}O_{carb}$ in the upper 303 cm of sediment range from -6.9 to -1.5‰ and -18.4
to -15.2‰, respectively, and the mean isotope compositions become more positive over the
record, but the long-term trend of $\delta^{18}O_{carb}$ is not statistically significant (Fig. 4). Variance in
$\delta^{13}C_{carb}$ and $\delta^{18}O_{carb}$ values is highest before 4.4 ka (below 200-cm depth) and lowest since 1.5 ka
(above 40-cm depth; Fig. 5). Isotope excursions appear in both the slow and fast sedimentation
intervals and when the carbonate flux is both low and high (Fig. 4).

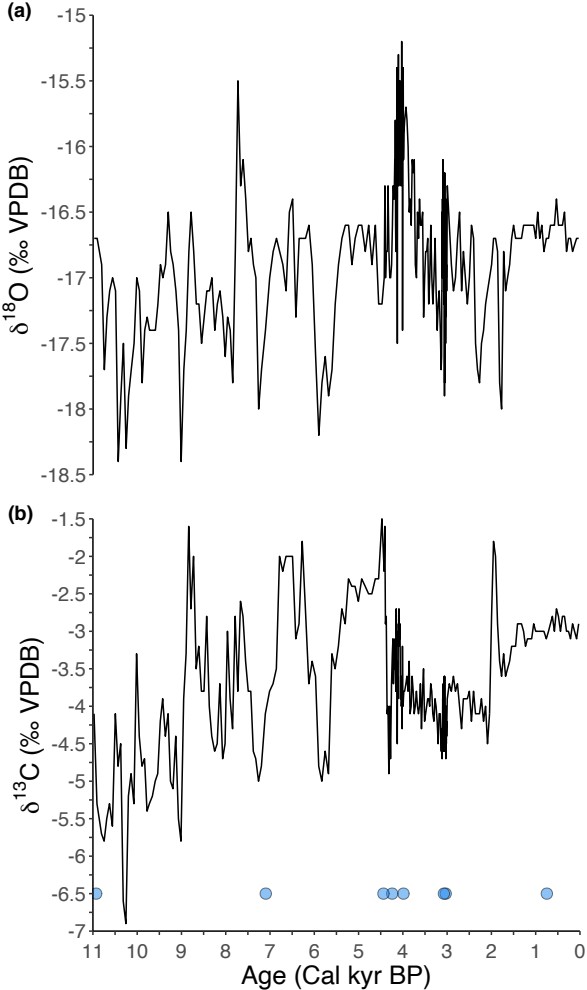


**Figure 5.** $\delta^{18}O_{carb}$ (a) and $\delta^{13}C_{carb}$ (b) from Highway 130 Lake. The blue dots indicate the calibrated

radiocarbon ages used for the age-depth model (refer to Table 1 for calibrated age uncertainties).

$\delta^{18}O_{carb}$ peaks from approximately 4.2–4 ka, where four calibrated radiocarbon ages

constrain the timing and indicate a fast sedimentation rate (Fig. 5). The carbonate flux is high,



but the carbonate content is low (~55%) during this interval relative to the mean (Fig. 4). A
positive excursion of similar magnitude also occurred from 7.8–7.3 ka, but aligns with high
organic content and low $\delta^{13}C_{carb}$, carbonate content, carbonate flux, and total net sediment
accumulation.
Compared to the records for Bison and Yellow Lakes in Colorado (Anderson, 2011, 2012; Fig.
1), $\delta^{18}O_{carb}$ values of HL are several per mil lower with higher variance for most of the Holocene
(Fig. 6). This pattern changes in the late Holocene as carbonate in Bison Lake becomes
isotopically lighter than before and approaches the oxygen isotope composition of HL, which
maintains a relatively constant mean $\delta^{18}O_{carb}$ value. After approximately 1.5 ka, $\delta^{18}O_{carb}$
variability in HL drops to near the analytical uncertainty (± 0.2‰) while the other records show
increased variably (Fig. 6).

5.  **Discussion**
*5.1 Evidence of the 4.2 ka drought in the southern Rocky Mountains*

Peak $\delta^{18}O_{carb}$ in HL indicates an abrupt decline in effective moisture or at least a decline

in the ratio of snowfall to rain in the Medicine Bow Mountains from approximately 4.2–4 ka
(Fig. 5) when evidence from additional climate records shows that aridity affected the southern
Rocky Mountains and portions of the Great Plains (Carter et al., 2013; Halfen & Johnson, 2013;
Stokes & Gaylord, 1993). The highest $\delta^{18}O_{carb}$ values at HL coincide with the pollen-inferred
precipitation and temperature changes at 4.2 ka at Long Lake, which records two centuries of
severe drought (Long Lake, Fig. 1; Carter et al., 2013). The excursion also aligns with the



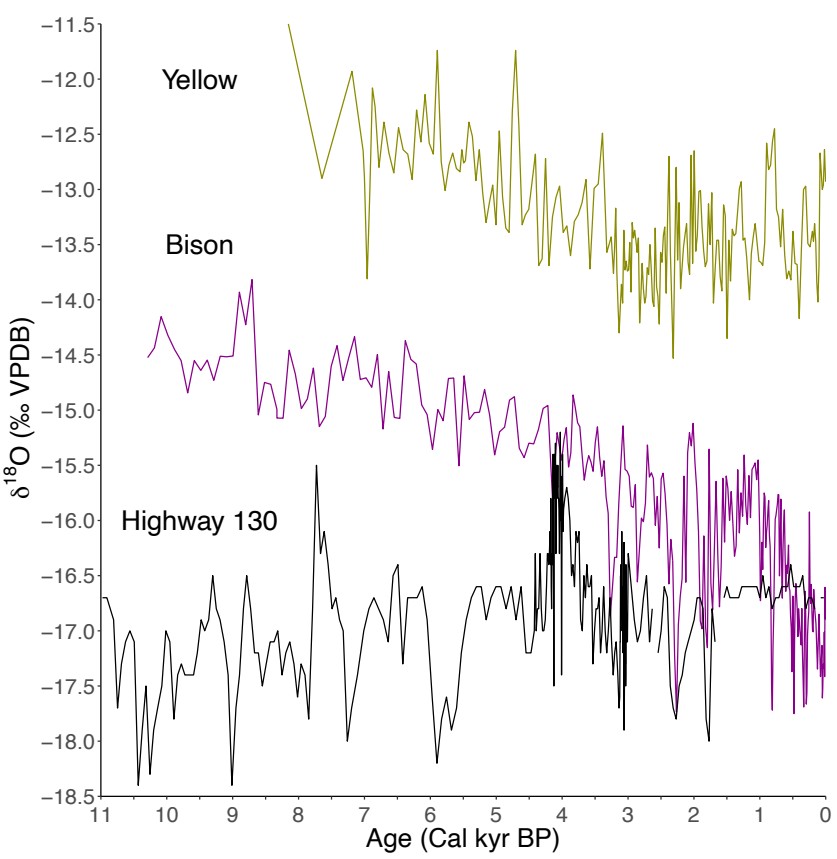

**Figure 6.** $\delta^{18}O_{carb}$ records from Highway 130 Lake (black), Bison Lake (purple; Anderson, 2011), and Yellow Lake (yellow; Anderson, 2012) vary despite their similar locations and elevations.

longstanding evidence of drought in the Great Plains and eastern southern Rocky Mountains
(dune fields, Fig. 1), where a rapid loss of grain-trapping vegetation likely triggered several
centuries of increased aeolian transport documented across multiple dune fields (Booth et al.,
2005; Forman et al., 2001; Halfen et al., 2010; Stokes & Gaylord, 1993).

Taken together, the records suggest that rapid drying at around 4.2 ka was an important

climatic event in the Medicine Bow Mountains even if the drought is not a prominent feature in
other paleoclimate studies from the mid-latitude Rocky Mountains (Anderson et al., 2008;



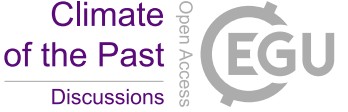

Brunelle et al., 2013; Feiler et al., 1997; Johnson et al., 2013; Mensing et al., 2012; Minckley et
al., 2012; Shuman et al., 2010; Thompson et al., 1993; Whitlock & Bartlein, 1993), including the
nearby $\delta^{18}O_{carb}$ records from Bison and Yellow Lakes (Anderson, 2011, 2012). The spatial
patterns of late-Holocene hydroclimate changes in North America may have been complex
compared to other regions, such as the European continent where late-Holocene climate
variability appears more coherently in climate records (e.g., Deininger et al., 2017). Still, the
inconsistent evidence complicates interpretations of the 4.2 ka anomaly here and elsewhere
(Bradley & Bakke, 2019).

Some paleohydrologic evidence indicates, however, that the event may have been

extensive in the southern Rocky Mountains. Sedimentological changes in high-elevation lakes in
Colorado show substantial hydrological transformation at around 4 ka matching the timing and
scale of drought inferred from HL's record (Fig. 1 & 7; Shuman et al., 2009a, 2014, 2015). The
sediment stratigraphies record low water levels that shifted shoreline sands to the locations of
cores collected in 1–5-m water depth today and thus indicate reduced effective moisture at 4.2–
3.9 ka (gray shaded regions, Fig. 7). The sand layers coincide in time with the elevated carbonate
accumulation rate and $\delta^{18}O_{carb}$ values at HL and with dune activity in southeast Wyoming
(Halfen & Johnson, 2013; Stokes & Gaylord, 1993); the high-elevation lake locations and
geophysical site surveys confirm that the shallow-water sands were not deposited by aeolian
activity. A second prominent sand layer in Emerald and Upper Big Creek Lakes at ca. 3.1 ka
(gray shaded regions, Fig. 7) indicates low effective moisture and overlaps with the maximum
rate of carbonate accumulation at HL, but a second sand layer does not appear in Little Molas
Lake and the $\delta^{18}O_{carb}$ values in HL are lower than at 4.2 ka. It took several centuries for $\delta^{18}O_{carb}$
values to rise and fall before and after the peak from 4.2–4 ka, but the excursion at 3.1 ka





occurred within a century. Multiple radiocarbon ages constrain the interval of high carbonate
accumulation from approximately 4.4–3 ka, but the sedimentation rate in the interval is sensitive
to removal of one of the ages; if the age at ca. 3 ka is out of sequence, we could bias the peak
rate of sediment and carbonate accumulation toward high values.

The rapid transition from deep-water muds to shallow-water sands as water levels

dropped in the Colorado lakes at around 4.2 ka corresponds with changes in pollen assemblages
in central Colorado (Jiménez-Moreno et al., 2019) and southeast Wyoming (Carter et al., 2013),
as well as with other evidence for drought in North America (Booth et al., 2004). Similar
sedimentological features found in lakes along the Atlantic margin from Maine to Pennsylvania
date to around 4.2 ka, for example, where the drought appears as one of multiple events linked to
circulation changes over the North Atlantic (Li et al., 2007; Marsicek et al., 2013; Newby et al.,
2014; Nolan, 2020; Shuman et al., 2019; Shuman & Burrell, 2017). The sequences in the
southern Rocky Mountains, however, include uniquely prominent isotopic and sedimentological
changes from 4.2–3.9 ka.

Given the growing evidence of drought within the southern Rocky Mountains associated

with the widespread climatic anomaly at 4.2 ka, a lack of $\delta^{18}O_{carb}$ records of the event in the
region, or in North America entirely, is surprising (Anderson et al., 2016b; Konecky et al., 2020).
However, individual sites respond to a varying mixture of local and regional factors. The
stratigraphic evidence of lake-level change in the region is not entirely consistent either and may
indicate interactions with different directions of hydroclimate change across seasons, elevations,
and latitudes. Stratigraphic features in Hidden Lake, located in northern Colorado just south of
Upper Big Creek Lake (Fig. 1) but several hundred meters lower in elevation, document a rapid
increase in effective moisture at around 4 ka (Shuman et al., 2009)—the opposite response of the

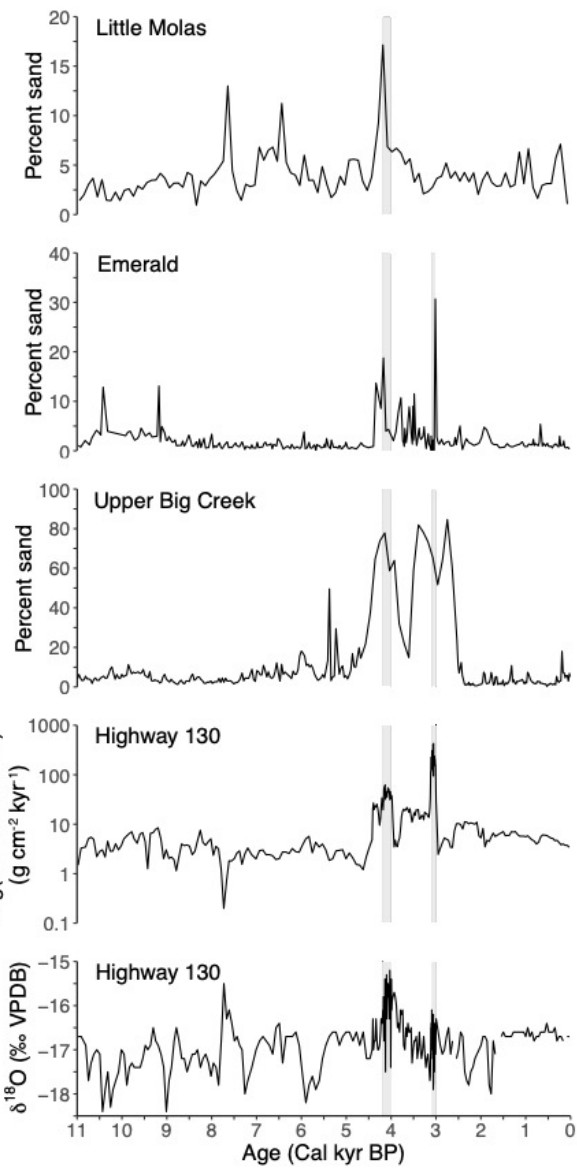


**Figure 7.** Spikes in the sand content of Little Molas, Emerald, and Upper Big Creek Lakes, located in high-elevation watersheds in Colorado (Fig. 1), align with the positive $\delta^{18}O_{carb}$ excursion at Highway 130 Lake and indicate low water from approximately 4.2–4 ka (gray shaded areas) resulting from low effective moisture (Shuman et al., 2009a, 2014, 2015). Another positive $\delta^{18}O_{carb}$ excursion at ca. 3.1 ka (gray shaded areas) aligns with intervals of low water at Emerald and Upper Big Creek Lakes.



surrounding lakes at higher elevations (Fig. 7). The wet phase was abrupt in onset and
termination and lasted from around 4.4–3.7 ka, which stands out in the otherwise gradual trend
towards higher water levels since 6 ka without any major intervals of low water in the late
Holocene (Shuman et al., 2009).
The low-elevation location of Hidden Lake may indicate an important role for increased
summer or fall rainfall when high-elevation sites declined in response to low winter snowfall.
The combined effects could have favored the unusually high $\delta^{18}O_{carb}$ at HL. Low winter snow
can create favorable surface-energy conditions for strong summer convective precipitation (Zhu
et al. 2005). Alternatively, the reversed hydrologic response of Hidden Lake could indicate
antiphased hydroclimate changes in the southern Rocky Mountains between high and low
elevations, which is consistent with modern responses to the El Niño-Southern Oscillation
(Preece et al., 2020). The active dune fields in east-central Wyoming, however, confound a
simple interpretation of the elevational and seasonally antiphased hydrologic changes.
Latitudinal hydroclimate variability could be an additional complicating factor and has
previously been described across the area due to transient climatic boundaries with the northern
and southern Rocky Mountains (Shinker, 2010; Wise, 2010). The comparison of the radiocarbon
age uncertainties of the 4.2 ka paleoshoreline sands at lake-level sites, including Emerald Lake in
central Colorado (Fig. 1 & 7), indicates a late-Holocene north-south moisture dipole extending
across much of the area described here (Shuman et al., 2014).
Given the potential prominence of the 4.2 ka drought at HL and other southern Rocky
Mountain records, it may have been uniquely severe in this region even if it had a complex
regional expression at broader spatial scales. The lake-level reconstructions from Colorado
contain evidence of other Holocene hydrologic changes (Fig. 7) and HL shows another positive





excursion at 7.8 ka (Fig. 5), but the records lack evidence for multiple recurrent, multi-century
hydroclimate changes recorded with the 4.2 ka event in places like the Atlantic margin (Shuman
et al., 2019). Elsewhere, aridity at 4.2 ka may represent just one of several repeated drying events
consistent with climate records and simulations from around the world that show drought as a
regular feature of late-Holocene climate variability (Arz et al., 2006; Bradley & Bakke, 2019;
Mayewski et al., 2004; Wanner et al., 2008; Wanner et al., 2015; Yan & Liu, 2019). The mid-
latitude Rocky Mountain records may suggest that the midcontinent was insulated from some of
the abrupt late-Holocene climate changes, possibly due to its isolation from the ocean-
atmosphere dynamics proposed to play key roles in Holocene variability (Arz et al., 2006;
Deininger et al., 2017; Jalali et al., 2019; Yan & Liu, 2019).

**5.2 Varying $\delta^{18}O_{carb}$ trends in the southern Rocky Mountains**
The marked sensitivity of lake-water $\delta^{18}O$ to hydroclimate changes may make lacustrine
carbonates ideal indicators of past droughts like the 4.2 ka event, as documented by $\delta^{18}O_{carb}$
records outside of North America (e.g., Bini et al., 2019; Dean et al., 2015) and by our record at
HL (Fig. 5), but site-specific hydrologic conditions could complicate the signals. They may
generate inconsistent trends among records over both short (seasonal) and long (millennial)
timescales (Gibson et al., 2016; Mark D. Shapley et al., 2008; Steinman & Abbott, 2013; Tyler et
al., 2007). Indeed, we observe such inconsistency in the southern Rocky Mountains (Fig. 6).
The hydrologic controls, such as groundwater fluxes and basin morphology, can vary
based on a lake's geohydrological setting (Anderson et al., 2016; Dean et al., 2015). Modern
lake-water hydrogen and oxygen isotope measurements reveal stronger fractionation by
evaporation in HL (thick black line, Fig. 2) compared to Bison and Yellow Lakes (purple and





yellow lines, Fig. 2), which exhibit a narrower range in modern water isotope values and smaller
deviation from the global meteoric water line (Anderson, 2012). However, the carbonate at HL is
isotopically lighter than the other two (Fig. 6), which is antithetical to the expectation based on
evaporatively enriched summer waters (Fig. 2). The pattern differs from the interpretation that
the Bison Lake $\delta^{18}O_{carb}$ was not strongly influenced by evaporation because it was isotopically
lighter than other sites like Yellow Lake (Anderson, 2011, 2012). HL lacks the prominent
$\delta^{18}O_{carb}$ trend observed at these other sites (Fig. 6). Given the modern water isotope values, we
had anticipated that $\delta^{18}O_{carb}$ from HL would be isotopically heavy compared to Bison and
Yellow Lakes, but track similar trends (Anderson, 2012).

Because the modern water isotope values poorly predicted $\delta^{18}O_{carb}$ trends, the different

lakes may record past changes in different ways. HL may be a better indicator of winter
snowpack than evaporation. $\delta^{18}O_{carb}$ values were below the mean at 6 ka (Fig. 5) when simulated
estimates of evaporation rates in the Medicine Bow Mountains were up to 30% higher than today
(Morrill et al., 2019), which would indicate that such enhanced summer evaporation did not
affect $\delta^{18}O_{carb}$ at HL. We also find lower-than-expected $\delta^{18}O_{carb}$ values in the uppermost
sediments. Despite an increase in summer lake-water $\delta^{18}O$ from -17.8 to -10.8‰ today (thick
black line, Fig. 2), $\delta^{18}O_{carb}$ values since 1.5 ka only reached a maximum of -16.4‰ and the core-
top value is -16.7‰ (Fig. 5), which is closer to the composition of groundwater (open squares,
Fig. 2) than the mid- to late-summer lake-water-$\delta^{18}O$ values (black circles, Fig. 2).

Previous studies have shown that the deposition of endogenic carbonate occurs

predominantly in the warm summer months when photosynthesis optimizes carbonate production
by modifying dissolved $CO_2$ concentrations and pH (Leng & Marshall, 2004), but the
isotopically light carbonate at HL may contradict this expectation. For comparison, the



uppermost $\delta^{18}O_{carb}$ value of -14.9‰ in Bison Lake (purple line, Fig. 6) falls within the range in
modern summer lake-water-$\delta^{18}O$ values of -16.7 to -13.5‰ (purple line, Fig. 2; Anderson,
2011). $\delta^{18}O_{carb}$ in HL, therefore, may not integrate the range of summer lake-water $\delta^{18}O$ as is
assumed for Bison and Yellow Lakes (and carbonate lakes in general), but early springtime
deposition of carbonate at HL could capture the signature of isotopically light lake water without
modification by warm-season evaporation.

The year-round measurements of specific conductance show that conditions favorable for

carbonate precipitation may indeed be highest during late winter and spring. In 2017, specific
conductance of the water below the surface ice rose from 700 µS/cm in early February to 1,115
µS/cm by early April, and it remained above 1,000 µS/cm throughout April (Fig. 3). These high
values would favor carbonate precipitation, whereas the summertime waters are more dilute.
Specific conductance of HL and other lakes within the watershed during the summer typically
does not exceed 300 µS/cm. Melting of lake ice and snowpack rapidly lowers the specific
conductance by early May and it remains between 250–300 µS/cm for the remaining ice-free
months. The conductance likely remains lower than in winter despite evaporative enrichment of
the oxygen isotopes because of groundwater discharge into the lake (Rautio & Korkka-Niemi,
2011), which geophysical surveys, water temperatures, and stable summer water levels at HL
support (Liefert et al., 2018). If so, ions exsolved from overlying ice raise the conductance of the
lake water and pore water within the bottom sediments beyond the concentration in groundwater
in winter (Adams & Lasenby, 1985), and create favorable conditions for the rapid deposition of
endogenic carbonate in early spring when the isotopic signal would not reflect evaporation or
isotopically heavy summer rainfall (open circles, Fig. 2). Spring carbonate formation could also
yield a different temperature-dependent effect on the $\delta^{18}O_{carb}$ in HL compared to the other lakes,

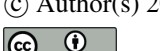



but the cold spring waters at HL should favor an increase, not decrease, in $\delta^{18}O_{carb.}$    Indeed, all
of the readily expected process that could complicate a carbonate isotopic record should drive the
$\delta^{18}O_{carb}$ in the positive (not negative) direction and underscore the significance of the difference
between HL and the other lakes.

As an alternative explanation, the $\delta^{18}O_{carb}$ values could reflect changes in total

precipitation rather than seasonality effects because the inflow of Ca-bearing groundwater
(which should rise with precipitation) can increase carbonate production and lower $\delta^{18}O_{carb}$
values in alkaline lakes in both ice-free and ice-covered conditions (Shapley et al., 2005), but the
weak covariance of weight percent carbonate and $\delta^{18}O_{carb}$ suggest that rates of groundwater
inflow did not strongly influence $\delta^{18}O_{carb}$ (Fig. S1a). A weak covariance of $\delta^{13}C_{carb}$ and $\delta^{18}O_{carb}$
indicates short lake-water residence times throughout the lake's history (Fig. S1b; Drummond et
al., 1995; Talbot & Kelts, 1990), which could be consistent with rapid flowthrough that reduced
evaporative enrichment; removing values from 4.2-4 ka only marginally improves the
correlation.

A strong negative correlation of weight percent organics and carbonate ($R^2 = -0.79$)

suggests that carbonate abundance depends primarily on biological productivity that promoted
carbonate dissolution by releasing $CO_2$ and lowering pH (Fig S1c; Dean, 1999). Carbonate
content from 4.2–4 ka was below the mean despite low organic content (red dots, Fig. S1c) and a
high flux of carbonate (Fig. 4), which may represent a shift in HL's water levels and chemistry
that favored both acidic conditions and isotopically heavy carbonate (red dots, Fig. S1). Down-
core shifts in $\delta^{18}O_{carb}$ produced by seasonal changes in the timing and rate of carbonate formation
have been proposed as potential sources of variability within individual records (Fronval et al.,
1995; Lamb et al., 2007; Steinman et al., 2012; Steinman & Abbott, 2013; Tyler et al., 2007) and



could play a role in the record at HL, but such differences could also generate the variability in
the long-term trends observed among records from the southern Rocky Mountains and elsewhere
(Bini et al., 2019; Konecky et al., 2020; Roberts et al., 2008).

Other factors that could affect $\delta^{18}O_{carb}$, such as precipitation patterns and biological

disequilibrium effects, are unlikely sources of variability among the regional $\delta^{18}O_{carb}$ records.
The seasonal balance of precipitation today is broadly similar among the sites (inset plot, Fig. 2)
and the calculated annual precipitation-$\delta^{18}O$ value is approximately 1‰ lower at HL
(http://waterisotopesDB.org). Annual temperature ranges are also similar for the watersheds,
making it unlikely that temperature dependence of fractionation could explain the range in
$\delta^{18}O_{carb}$ values recorded across the three records unless the different water depths and
groundwater influences altered the seasonal temperature progression among lakes. The
difference in temperature would need to be large (~12 °C) to explain the offset in $\delta^{18}O_{carb}$
between HL and Bison Lake (and larger for the offset between HL and Yellow Lake), which is
unrealistic given the sites' comparable locations and elevations and the relatively small
temperature changes at mid-latitudes since 11 ka (Marsicek et al., 2018). We also find no
evidence in the sediment core or modern lake setting to indicate that biologically mediated
precipitation of calcite substantially altered $\delta^{18}O_{carb}$ at HL (e.g., by the accumulation of ostracod
tests that precipitate carbonates in disequilibrium with lake water). Disequilibrium effects
associated with biogenic carbonates generally increase $\delta^{18}O_{carb}$ (Holmes & Chivas, 2002; Leng &
Marshall, 2004), which would be difficult to reconcile with the surprisingly negative mean and
core-top $\delta^{18}O_{carb}$ values at HL. Down-core carbonate phase changes are also unlikely as we
identified that only calcite was present using x-ray diffraction (XRD).



6. **Conclusions**
$\delta^{18}O_{carb}$ from HL indicates an abrupt hydroclimate change in the southern Rocky
Mountains from approximately 4.2–4 ka that reduced effective moisture or caused less snow to
fall than today at high elevations in southern Wyoming. Other $\delta^{18}O_{carb}$ records from the region
do not document the drought (Fig. 6; Anderson, 2012), but the event's timing overlaps with
evidence of multi-century drought from pollen, lake stratigraphies, and dunes in the southern
Rocky Mountains (Carter et al., 2013; Halfen & Johnson, 2013; Shuman et al., 2009a, 2014,
2015; Stokes & Gaylord, 1993), the western Great Plains (Booth et al., 2005; Dean, 1997; Halfen
& Johnson, 2013; Mason et al., 1997; Stokes & Gaylord, 1993), and elsewhere around the world
(Nakamura et al., 2016; Di Rita & Magri, 2019; Scuderi et al., 2019; Xiao et al., 2018).
The timing and magnitude of hydroclimate change in our record agrees with the
perspective of a widespread megadrought at around 4.2 ka (Weiss, 2016), but inconsistencies
among climate records suggests that (1) site-specific factors can prevent identification of the
patterns of abrupt hydroclimate changes, particularly in $\delta^{18}O_{carb}$ records; (2) the hydrologic
response in North America and likely elsewhere around the world was spatially complex; and (3)
the abrupt hydroclimate changes in the North American midcontinent were more pronounced
against background Holocene variability than in many regions such as the Atlantic margin.
Consequently, a prolonged 'megadrought' at 4.2 ka was likely a significant feature of the
hydroclimate history in the mid-latitude Rocky Mountains even if that is not true globally.

**Data availability**
Data related to this paper will be made available through the National Centers for Environmental
Information on the National Oceanic and Atmospheric Administration website:



https://www.ncdc.noaa.gov/data-access/paleoclimatology-data. The analyses were performed in
R.
**Author contributions**
D. Liefert and B. Shuman contributed to the design and implementation of the research, to the
analysis of the results, and to the writing of the manuscript.
**Acknowledgments**
This project was funded by the National Geographic Society (CP-064ER-17), U.S. National
Science Foundation P2C2 (EAR-1903729), the Wyoming Center for Environmental Hydrology
and Geophysics via support from the U.S. NSF EPSCoR program (EPS-1208909), and the
Department of Geology and Geophysics at the University of Wyoming. We thank Andrew
Parsekian and Kevin Befus for field assistance and Andrew Flaim for assisting in sample
preparation.

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
