# Peer review of "Expression of the "4.2 ka event" drought in the southern Rocky Mountains, USA"

_Climate of the Past, 2021_

## Author Response (AR1)

**Response to Reviewer 1**

*Obviously your interpretation rests on δ18O being weighted more towards the spring. You do justify this well using the conductivity data, δ18O-d13C covariance, etc. Ideally you would have used sediment traps to establish when most carbonate is deposited in a year – maybe something to think about if you continue your work on this lake as then you'd be able to know with more certainty when carbonate precipitated.*

**Sediment traps would indeed help to clarify the timing of carbonate production in the lake and aid in interpreting the isotope record. Installing sediment traps over winter would be challenging because there is very little liquid water between the lake bed and overlying ice, and some carbonate production could occur within the bottom sediments rather than water column. However, measuring sediment in summer would be straightforward and informative. We added text to suggest these measurements in future work, as indicated below.**

*As I say above, if the d18O is just influenced by snowpack change, is "drought" really the best word to describe the 4.2ka event here? But anyway, clear some hydroclimate change going on, which is useful to know.*

**We agree that referring to the climate event described here as "hydroclimate change" would be accurate and broadens the scope of potential environmental changes that had affected our record, such as a reduction in snowpack. Because reduced snowpack is a type of drought (often referred to as a "snow drought"), the term "drought" also accurately portrays the type of hydroclimate change indicated by our record. Describing the event as a drought is also consistent with the interpretations and description of the nearby pollen-derived climate record in the Medicine Bow Mountains, WY, by Carter et al., 2018. To be transparent about our interpretations of the record, we have dropped the word "drought" from the title of the manuscript but continue to reference droughts throughout the text where it is supported by analyses of the specific hydroclimate changes and publications with similar conclusions.**

*I'm not really sure why you have plotted the age model and the LOI on the same graph.*

**Overlaying the age model and LOI helps to illustrate the relationships of sediment accumulation and carbonate production. For example, high rates of net sedimentation correspond with intervals of high carbonate flux into the lake, indicating that carbonate production may largely control sedimentation rates. The data shown in the first panel of Fig. 4 helps to clarify that this relationship is not true for carbonate content, which was both high and low through the fast-sedimentation interval. The age-depth model also controls the inferred timing of sediment and isotope changes, and overlaying the data helps to show that the model is well constrained within positive isotope excursion centered at around 4 ka when carbonate values were low but the carbonate flux was high.**

You say "Ostracod tests were present in less than 10 of the 300 samples." Obviously these could have different δ18O to endogenic calcite. Can you just briefly confirm that these aren't all around 4.2ka or something, to check they aren't responsible for the excursion at that time.

**Ostracods and other materials were noted during sample preparation and later confirmed to not be found in this section of core. They are also unlikely to explain trends in the data because the sedimentation rate was very high at around 4.2ka, meaning that nearly 0.5 m of sediment accumulated during the positive isotope excursion, far more material than had contained ostracod tests throughout the entire length of the core (3.3 m).**

Line 382 – just Shipley et al., 2008

**Line 509: The citation has been corrected.**

**Line 661: We added the text "Installing sediment traps during the ice-free season could clarify the timing of carbonate production."**

**Response to Reviewer 2**

Regarding the Abstract/Introduction and general framing: I found the description of the '4.2 ka event' a little confusing. The authors switch between describing an event at 4 vs 4.2 ka (are these the same thing?), and also provide quite vague background about the global nature of the event compared to the greater detail provided for North America (including potential forcings). I'd suggest just a sentence or two about the event in a global context before jumping more immediately into discussing the event in the context of North American climate (which, after all, is the focus of this paper)

**A sentence has been added at line 67 to expand the discussion of possible drivers of the event globally.**

It would be nice to see some discussion of possible anthropogenic influences on this (and other) lakes, even if it's just a referenced statement like 'there was probably not any anthropogenic influence'

**Text was added to line 635 explaining that we have no reason to expect anthropogenic influence.**

Line 26: define 'ka' at the first instance

**The suggested change has been made (line 30).**

Line 28: abrupt **global** drying?

**The sentence was revised to indicate that evidence of drying exists primarily in the Northern Hemisphere (line 30).**

Line 35: "…records from Colorado do not record it." – what exactly is 'it'? we've lost the subject that this 'it' should be attached to

**"it" has been changed to "drought" (line 36).**

Line 40: 'the strong enrichment…..summer months today' I suspect that this sentence may be referencing an erroneous comparison of lake water $\delta^{18}O$ and lake carbonate $\delta^{18}O$ values that I point out later on. If so, this should be removed.

**Text was added to line 248 describing how we calculated temperature-dependent fractionation of calcite formation and conversion of VSMOW to VPDB. See responses to comments below for more changes.**

Line 45: 4 ka and not 4.2 ka? Is this meant to be the same 'event'?

**Correct. Records from around the world interpreted to support the "4.2 ka event" span multiple hundreds of years around 4 ka and many do not begin or are not centered directly on 4.2 ka. On Line 30 we now reference this point.**

Line 50: list dates (in parentheses) of the YD chronozone as a reminder for us

**The suggested change has been made (line 53).**

Line 59-61: This sentence is a bit grammatically ambiguous; I suggest rearranging it along the lines of 'However, some regions show increased precipitation, which is consistent with…"

**Done (line 62).**

Line 62: 'Recent' -> 'Recent **model**'

**The suggested change has been made (line 70).**

Line 67: Unless I'm mis-remembering, Ault et al 2018 specifically describes drought in western North America (i.e. this isn't globally applicable). In any case, I suggest that by here you have already focused in on the nature of the '4.2 event' in North America (not globally)

**Ault et al. indeed describes dynamics in western NA, but their finding that abrupt climate changes can occur from intrinsic climate variability (as opposed to some external forcing) is relevant to 4.2 ka studies globally and provides context for distinguishing the event from other Holocene variability, much of which was driven by external processes. By not changing or removing this paragraph to focus on North America, it provides the background on the event in a global context requested in the first bullet point of this review. No change was made.**

Line 75-76: put the 'in the North American midcontinent' modifier earlier in the sentence; this is grammatically ambiguous as written

**The sentence has been separated into two for clarity (line 92).**

Line 80: I suggest putting the 'However' at the start of this sentence for clarity

**Done (line 94).**

Line 84: what exactly is a 'dune record'? Is this a 'dune-field chronology' as per below? If yes, you should write that out here too

**Done (line 98).**

Line 83: 'Rocky Mountains **of North America**'

**Done (line 97).**

Line 85: It would be good if here you also listed the proxy record types that don't show evidence for a 4.2 ka event

**Stable isotopes are the best example here (line 99).**

Line 94-97: Two 'prominent's in one sentence (just in case you want to change one)

**The first instance was removed (line 113).**

Line 97 (last word): again, what is 'It'?

**The sentence was clarified to indicate a drying event (line 116).**

Line 100: 'By contrast, the 4.2 ka…' -> 'By contrast, **a** 4.2 ka…'

**The existing text is accurate because the 4.2 ka event is thought to represent a single climatic anomaly identifiable from multiple sources rather than one of many 4.2 ka events. No change was made.**

Line 117: measurements of what? Something like 'Measurements of modern lake water physical and geochemical characteristics can help…' might be clearer

**The sentence was clarified to indicate isotopic measurements (line 147).**

Line 177: controls on what? Lake carbonate $\delta^{18}O$? Lake water $\delta^{18}O$? Other?

**This is likely referencing line 117, not 177. Carbonate d18O was added for clarification.**

Line 121: You could reference Figure 1 here

**The suggested change was made (line 154).**

Line 127: spell out 'water isotopes' at the first instance i.e. 'water stable isotopic compositions ('water isotopes' hereafter)'

**Done (line 160).**

Line 132: 'interpretations of **the stable isotopic composition of lacustrine carbonate interpreted in terms of past hydroclimate variability**' or similar

**The sentence has been revised similarly (line 170).**

Line 142: 'but high elevations' -> 'but high-**elevation sites**'

**The sentence was reduced for clarity (line 185).**

Line 145: could you just say 'average annual temperature range'?

**The existing sentence describes the annual temperature extremes rather than average range. No change was made.**

Line 146: add reference

**Done (line 188).**

Line 163: were these precipitation/groundwater samples collected at the same time/over the same time interval as the lake water samples? Either way, you should state the collection dates.

**Done (lines 207 and 209).**

Figure 1: Add a spatial scale of some sort to inset a (eg lat/lon). It would also be good to highlight Bison & Yellow lakes in some way, given you do a lot of explicit comparison of your new observations with similar observations from these lakes. Additionally, could you not slightly extend box b so that it includes Little Molas Lake? It would be good to be able to see it, given you show data from this lake in Figure 7 and it's a bit odd that it's the only lake cut out.

**The suggested changes were made (Fig. 1).**

Line 164: 'Isotopic ratios **of all water samples** were measured…'

**Done (line 210).**

Line 166: Here (or at least somewhere) you should state that water stable isotopic ratios are reported relative to VSMOW (this is an important distinction from your carbonate values, for which you do state the standard)

**Text has been added to this paragraph to indicate the water standard (line 213).**

Line 182: 'At the same time' at the same time as what, exactly? Better just to state the time again (I am guessing January 2017, in which case something like 'In January 2017, we also collected…')

**Done (line 231).**

Line 186-line 189: Your methodology here is a bit unclear. Do you mean to say that you roasted the samples at 550 degrees, then performed stable isotopic analysis on the carbonate from that roasted sediment? What are the oxidizing agents mentioned in line 188? Did you oxidise the roasted/raw sediment, or just the roasted sediment? It would also be good to show the results of this comparison (mentioned in line 188) as a supplementary figure

**This sentence has been revised for clarity (line 236).**

Line 189-190: grammatically ambiguous; I think you mean to say that you sieved out the fine fraction, and then measured the stable isotopic composition of that fine fraction using the mass spec?

**Correct. The sentence has been revised for clarity (line 239).**

Line 192: if the calcite isn't ostracod tests, then what is it? Amorphous fine-grained? Unidentified but probably autochthonous? Do you have any SEM (or other microscope) images of this carbonate? It would help the reader a LOT throughout the rest of the paper to have at least some idea of the nature of this lake carbonate

**In this sentence and the next we elaborated on the type of calcite present and our reasoning (line 243).**

Line 195-198: I don't really understand what you are trying to say in this sentence; consider re-writing into several shorter sentences each describing one thing. Also you state here that you isolated conifer needles, but I don't see them on Table 1(?)

**The sentence was clarified and split into two. Conifer needles were referenced in error and have been removed (line 262).**

Line 204 and all later instances where you report stable isotopic compositions of lake **water**: I assume that these values are relative to VSMOW, which is an important distinction from your lake carbonate $\delta^{18}O$ values which are reported relative to VPDB. These two things are **not directly comparable in terms of their absolute values**

**This point is addressed in other comments, but here the text only refers to the composition of lake water, so no change was made.**

Line 205: unless I am mistaken, the 'thick black line' on Figure 2 is the LEL defined by your samples, but also shows the range in values (comparable to the arrows for the other lakes)? I found this a bit confusing so probably other readers will as well. Maybe re-think how you show the various data on this figure.

**Correct, this line shows both the range in samples and slope of the local evaporation line. The sentence was revised to explain more clearly that the slope of the line tracing HL's range in lake water isotope values (which define HL's local evaporation line) follow the local evap line of lakes in the CO Front Range (line 272).**

Line 207: 'Several consecutive years'?? Where are these data from? In the methods, you mention only that you collected lake water samples in 2017.

**The methods (line 209) were corrected to indicate the range of sampling dates (2015–2017). A sentence was added to Fig. 2's caption to indicate the data shown are only from 2017.**

Line 208: '**water** isotope values at HL'

**Done (line 275).**

Line 214-215: Are the water isotope values from these lakes truly comparable in terms of absolute range of variability? Do the measurements represent approximately the same seasonal range/duration of collection?

**The preceding sentence (now line 280) was revised to indicate the months and approximate year when Anderson collected these samples, which was earlier than 2017 (Anderson doesn't provide the exact year of when they were collected) but represents the same seasonal range as sample from HL.**

Line 218: Actually, just eyeballing the inset plot in Figure 2, it looks like the snow/rain ratios at the two lakes were quite different in 2017 when your data were collected

**This sentence was revised to clarify that it's the long-term average conditions that are similar rather than specific years. The average conditions are more important than individual years because carbonate oxygen isotope values in these lakes are integrated over decades (line 295).**

Line 222: Provide a reference for the lake-water temperature range at HL

**Text was added to lines 206 and 219 in the methods to indicate that water temperatures were measured with lake water samples and concurrently with depth using the pressure transducer.**

Line 231: Add a citation at the end of this sentence

**Done (line 308).**

Figure 2: from what data were the dotted LELs calculated? You should put the references explicitly in the figure caption. Also for ease of reading, at the filled black dots and thick black line to the figure legend

**The references are in the second sentence of the caption but were moved up in the sentence for ease of reading. The legend was revised (Fig. 2).**

Line 239: Remove both instances of 'in' after the percentages

**Done (line 316).**

Line 244: Here is another instance where I'd really like to know already how the carbonate is being produced in this particular lake!

**"Authigenic" was added for clarification (line 324).**

Figure 4 (and also Figure 5): It would be better if you combined these two figures, by simply plotting all the timeseries from Figure 4 on a **time axis**, and then showing the age-depth model as a supplementary figure (along with the core image, which doesn't add a huge amount given how narrowly it is shown). That would make later comparisons of these timeseries much easier. You could also then highlight time windows of interest.

It would also be much better (and would aid in some later interpretation) to follow modern best practice & incorporate the chronological uncertainty into your plotted timeseries (which are currently shown on only one realisation of the age-depth model) – there are many examples of this in recent palaeoclimate literature, as well as guides as how to do such things (e.g. the recently-published geoChronR package from McKay et al).

**Uncertainty bands have been added to Figures 5, 6, and 7 using geoChronR. Figure 4 was not changed so that it is clear to the reader how the raw data and age-depth model were used to generate the $\delta^{18}O$ time series in the following figure, particularly as it relates to the changes around 4.2 ka during a high rate of sediment accumulation (which isn't apparent if we plotted all the data on a time axis). The geoChronR results place low confidence in the other positive excursions (~8 and 3 ka) in the record besides at 4.2 ka. We have therefore removed text discussing these excursions and modified Fig. 7 by removing the gray shaded region at 3 ka.**

Line 259-260: might as well just say 'there is no significant trend'

**Done (line 341).**

Line 262: are these 'isotope excursions' statistically significant? That is, did you define them quantitatively in some way? Or are you just eyeballing peaks? If the former, you should describe the method that you use to identify anomalous intervals. If the latter, then you should either attempt some quantitative analysis, or say explicitly that the 'excursions' are qualitative.

**The excursions are now defined as deviations from the mean (e.g., the excursion at 4.2 ka represents a departure from the mean of three standard deviations) (line 346).**

Line 282: You need to define how exactly a change in the ratio of snowfall to rain manifests as a change in lake carbonate $\delta^{18}O$.

**Two sentences were added here to expand this discussion (line 386).**

Figure 6: Consider plotting these three records on their own y-axes. This would make the plot a lot clearer, and also the absolute values are not really of value here, but rather the variability

**Done (Fig. 6).**

Line 292-293: are 'the records' mentioned here all in the Medicine Bow Mountains? Throughout the discussion I lose track of which records do versus do not have evidence for a climatic anomaly at 4.2 ka, and also where they are (Medicine Bow Mountains, other parts of the Rockies etc). This could be quite easily clarified via a **table** (probably near Figure 1), listing the names of each site that you mention in the text, the proxy type, the region name, and whether or not there is evidence for some sort of event around 4.2 ka (and what that event was – drying, warmth other etc).

**Done, see Table 1.**

Line 304: 'high-elevation lakes' – there are only a few that you are referring to, so it would be clearer for the reader if you listed them by name

**The suggested change has been made.**

Line 307: 'the sediment stratigraphies **in these three lakes**'

**Done (line 409).**

Line 320: is there reason to suspect that this age is out of sequence? If so, this should be mentioned in the results. This potential bias from the age-depth model could also be addressed by showing age uncertainty on you plots as I suggest above

**It is already mentioned on line 328.**

Line 330: From what you have plotted here, in most cases the sedimentological changes at 4.2 ka do indeed look unique, but I wouldn't say that that is the case for the isotopic values

**We agree. The sentence has been reworded accordingly (line 452).**

Line 333: 'associated with the widespread climatic anomaly'- this is the hypothesis you're testing here, so you can't really cite it as being associated with the widespread North American drought (which is also something that you are assessing!)

**This text has been removed as suggested (line 454).**

Line 347: 'when **precipitation at** high-elevation sites…'

**The sentence was clarified, "sites" in this context referred to high-elevation lakes and their water-level declines (line 470).**

Line 348: How, exactly would these changes result in high lake carbonate $\delta^{18}O$? Some known influence on precipitation $\delta^{18}O$, which is then passed on to the lake carbonate $\delta^{18}O$?

**The sentence was expanded to help clarify this point (line 472).**

Line 361: 'Given the potential prominence of the 4.2 ka drought at HL': I'm still not exactly convinced of a mechanism linking the high lake carbonate $\delta^{18}O$ values and local drought conditions

**This discussion was revised to clarify our reasoning, see the two paragraphs starting on line 530.**

Line 376: This section might be better off at the start of the discussion – that way the reader has been introduced to the possible drivers of carbonate $\delta^{18}O$ values in the various lakes, the climatic implications of which can then be placed into the wider context

**This section of the discussion has been revised to frontload some of these concepts. We did not move it to the front of the discussion so that the hydroclimatic implications remain the emphasis rather than the possible controls on isotope records.**

Line 390: I am not convinced that there is much worth in comparing the absolute magnitude of carbonate $\delta^{18}O$ values from different lakes, especially given how far they are apart. There are WAY too many processes (climatic and otherwise) that can affect absolute values, even if there are common drivers of variability

**This discussion was revised to clarify our reasoning, see the two paragraphs starting on line 530.**

Line 398: So increased lake carbonate $\delta^{18}O$ at HL indicates less snowpack? Why, exactly? I think that you allude to various possible reasons but you should clearly outline the connection in terms of water isotope systematics.

**This paragraph and the one following it were combined and substantially revised to clarify our reasoning (line 530).**

Lines 403-406 and 411-416: Unless I am mistaken, here you seem to be directly comparing the absolute values of lake water $\delta^{18}O$ (relative to VSMOW) and lake carbonate $\delta^{18}O$ (relative to VPBD). This is not valid. Even when autochthonous lake carbonate precipitates using lake water as its source water, the fractionation depends on various things including the temperature at the point of carbonate precipitation (this is an unknown, in your case). Any conclusions that you have drawn based on comparison of absolute lake water and lake carbonate $\delta^{18}O$ values should either be removed, or re-thought in the context of anomalies.

**Text was added to the results (line 359) explaining how carbonate $\delta^{18}O$ was calculated using lake-water $\delta^{18}O$ and the range in potential lake temperatures. Lines 540 and 639 were revised to include the range in core-top carbonate $\delta^{18}O$ calculated from lake water $\delta^{18}O$ and the measured lake water temperatures. The reference to Bison Lake's values was removed.**

Line 417-421: this information would have been nice to know much earlier on – you could possibly sneak it into the results when you outline the specific conductance (or at least when you first discuss result from HL).

**This information is already in the Results beginning on line 303.**

Line 463: 'approximately 1% lower at HL' what exactly is lower than what?

**The sentence was revised (line 643).**

Paragraph starting line 460: The premise of this paragraph seems a little flawed to me. Again, discussing difference in absolute magnitudes of lake carbonate $\delta^{18}O$ between these three lakes is not particularly valuable, given the huge range of things (carbonate phase, seasonality, precipitation regime, seasonal cycle of precipitation $\delta^{18}O$, groundwater input, groundwater $\delta^{18}O$, local geology………) which could affect these absolute values, and which you don't have enough information to tease out. It's a comparison of variability(trends and other features of the timeseries) which is interesting (and relevant)

**This paragraph has been revised to directly address the points raised here and to expand on our reasoning for discussing both the magnitude and range of $\delta^{18}O$ (beginning on line 630).**

Line 476: what are they 'surprisingly' negative?

**The sentence was expanded for clarity (line 637).**